# Effect of Protection of Mountainous Vegetation against Over-Grazing and Over-Cutting in South Sinai, Egypt

**Kamal H. Shaltout** [1], **Ebrahem M. Eid** [2,3,*], **Yassin M. Al-Sodany** [3], **Selim Z. Heneidy** [4], **Salma K. Shaltout** [1] **and Safaa A. El-Masry** [1]

1   Botany Department, Faculty of Science, Tanta University, Tanta 31527, Egypt; kamal.shaltout@science.tanta.edu.eg (K.H.S.); salma.shaltout@science.tanta.edu.eg (S.K.S.); safaa.elmasri@science.tanta.edu.eg (S.A.E.-M.)
2   Biology Department, College of Science, King Khalid University, Abha 61321, Saudi Arabia
3   Botany Department, Faculty of Science, Kafrelsheikh University, Kafr El-Sheikh 33516, Egypt; yalsodany@sci.kfs.edu.eg
4   Botany and Microbiology Department, Faculty of Science, Moharam Bey, Alexandria University, Alexandria 21511, Egypt; selim.heneidy@alexu.edu.eg
*   Correspondence: eeid@kku.edu.sa or ebrahem.eid@sci.kfs.edu.eg; Tel.: +966-552717026

**Abstract:** In this study, we evaluated the species diversity, density, cover, and size index of plant species within and outside 37 enclosures in the South Sinai mountainous region (Egypt), which had been protected for six years (March 2012–March 2018) against over-grazing and over-cutting for medicinal and fuel purposes. Within and outside the enclosures, the plant species were recorded, and their density (individuals per 100 m$^2$) and cover (cm per 100 cm) were estimated using the line-intercept method. The biovolume of each individual of each species was calculated as the average of its height and diameter. The species richness was calculated as the average number of species per enclosure, and the species turnover was calculated as the ratio between the total number of species and the species richness. The relative evenness was calculated using the Shannon–Weaver index, whereas the relative concentration of dominance was calculated using the Simpson index. Detrended correspondence analysis (DCA) was applied to ordinate the vegetation inside and outside the enclosures depending on the species cover. The unpaired *t*-test was applied to assess the statistically significant differences in the species density, cover, and biovolume inside and outside the enclosures. By the end of the six-year period, the vegetation pattern inside the enclosures became more or less stable, presumably because of the stopping of grazing and cutting, which also led to an increase in the plant diversity, density, and cover. In general, the protection of vegetation in South Sinai improved its diversity, density, and cover. In addition, the topographic and physiographic heterogeneity in this region results in microclimatic variations, which play a major role in governing its natural vegetation.

**Keywords:** enclosure; MacArthur species distribution; microrefugia; regeneration of vegetation; Saint Katherine; species diversity

## 1. Introduction

The establishment of enclosures is a common rangeland rehabilitation strategy in semi-arid areas. Wairore et al., [1], in their study in west Pokot County (Kenya), reported that the enclosures have the potential of contributing to the resilience of vegetation in this region. Rong et al., [2] in the Junggar Basin (China), reported that excluding sheep grazing from a desert steppe for eight years increased plant cover and approximately tripled the biomass of the standing vegetation, particularly the shrub component. On the other hand, the diversity components as measured by the Simpson and Shannon–Wiener indices did not differ between the grazed and ungrazed areas. In addition, the study of Teketay et al., [3] on the woodland in northern Botswana recorded that the enclosure had a seven-times higher mean

density of woody species compared to outside of it, with exceptional regeneration of the seedlings inside. In the highlands of Tigray (Northern Ethiopia), Gebremedihin et al. [4] reported that the restoration of degraded drylands through enclosures enhancing the woody species diversity, soil nutrient diversity, and species richness was higher in the enclosures compared to in grazing lands.

South Sinai harbors the Saint Katherine Protectorate (SKP), which is one of Egypt's largest protected areas and includes the country's highest mountains. This arid, mountainous ecosystem supports surprising biodiversity and a high proportion of endemic plants [5]. Recent development pressure, catalyzed by tourism, has resulted in over-exploitation of the natural resources of this fragile region, particularly the plant resources. A management plan by the Egyptian Government is under development to conserve the area's natural resources and to ensure community participation while expanding opportunities for sustainable tourism. Local community guards now enforce a conservation plan in these spectacular and fragile regions [6].

The mountains of South Sinai have been one of the important centers of plant diversity for the Irano-Turanian region [7]. Approximately 472 plant species are found in the SKP; of these, 19 species are endemic, and more than 115 have known medicinal properties and are used in traditional therapies and remedies [5,8]. Local Bedouins use these species to treat various medical disorders, ranging from colds, digestive problems, and skin disorders, to bites and stings. Several species have properties that have attracted international medical interest (e.g., *Cleome droserifolia* is being investigated pharmaceutically for the treatment of diabetes [9]).

Shaltout and his team have carried out two studies on the vegetation of South Sinai. The first [10] aimed to assess the species diversity, soil, and water characteristics and the suitability of 26 Bedouin farms in the SKP for the cultivation of wild medicinal plants. The second [11] aimed at assessing the vegetation in wadi beds in this region. The wadi beds were analyzed in terms of the species composition, abundance, life forms, national and global distribution, depiction of the prevailing plant communities, and assessment of the role of the edaphic conditions that drive wadi bed vegetation. The present study aimed to evaluate the role of enclosures as a conservation tool for the protection of vegetation in this region. Our hypothesis was that there would be differences in the vegetation descriptors (diversity, density, cover, and biovolume) between the set of 37 fenced and non-fenced plots after six years of protection (March 2012–March 2018) against animal grazing and cutting for medicinal and fuel uses.

## 2. Materials and Methods

### 2.1. Study Area

The SKP was established in 1996 by the Egyptian Environmental Affairs Agency (EEAA) to protect the entire massif of the high mountains in South Sinai [6]. This massif is situated in the southern part of Sinai and is a part of the upper Sinai massif (33°55′ to 34°30′ E and 28°30′ to 28°35′ N). The natural conditions and geographical position of Sinai, as a bridge between Asia and Africa, have made it a distinctive biological region with its own characteristic flora and fauna (Figure 1). Floristically, it belongs to the Saharo-Arabian phytogeographical region (a subcategory of the Irano-Turanian chorotype). Its surprising biodiversity includes a high proportion of endemic and rare plants [12].

Mountainous flora in South Sinai differ from those in other areas because of the region's unique geology, morphology, and climatic aspects. The soil is formed mainly from the weathering of mountains that are mainly of granitic origin. The soil layer is generally shallow where the bedrock is close to the surface. Drought, over-grazing, over-harvesting, intensive tourism, urbanization and settlement expansion, unmanaged scientific research, and quarries have been reported as the main threats to plants [6].

The region is extremely arid with long, hot, rainless summers and cold, rainy winters and lies in the low rain belt of Egypt with an annual rainfall of 57 mm/year. However, its high mountains receive higher amounts of precipitation (100 mm/year) as rain and

sometimes snow. Nonetheless, rainfall is not an annual characteristic; rather, two to three consecutive years without rainfall is common. Rain takes the form of sporadic flash floods or limited local showers; thus, high spatial heterogeneity in receiving moisture is also common.

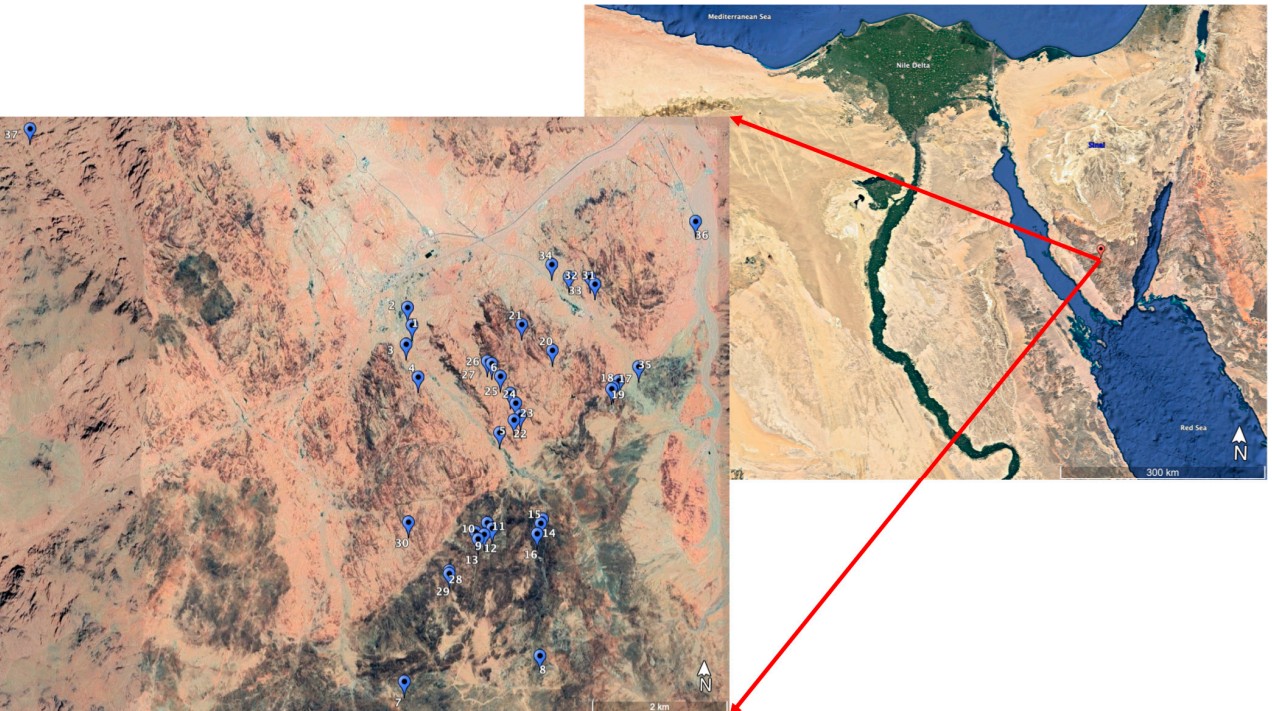

**Figure 1.** Location map of the 37 enclosures (1–37) located in the South Sinai mountainous region, Saint Katherine, Egypt (prepared from Google Earth).

### 2.2. Enclosure Description

Thirty-seven permanent enclosures of various areas, ranging between 6 m$^2$ (No. 27) and 246 m$^2$ (No. 21), were established to protect and monitor the threatened plant species in the SKP (Table 1, Figure 1). The size of the enclosures was different because the topography in this region is heterogeneous, and, in many cases, we could not choose comparable areas. The area of the enclosures was relatively estimated as a proportion of the a priori diversity of each site.

These enclosures represent eight habitats as follows: 17 along the wadi slopes; 7 in wadi beds; 6 in gorges; 2 in plains, terraces, and caves; and 1 in the foothills. Their altitudes range between 1533 m above sea level (No. 36) and 2377 m above sea level (No. 28). Nine enclosures are characterized by *Seriphidium herba-alba*, five by *Origanum syriacum* subsp. *sinaicum*, three by each of *Tanacetum sinaicum* and *Phlomis aurea*, and two by each of *Echinops spinosus* and *Artemisia judaica*. The other 13 enclosures are characterized by different species (Table 1).

The enclosures were selected to represent, as much as possible, the environmental variations associated with the distribution of the target species and were protected against animal grazing and human activities by fencing. The fences were intended to be permanent for the purpose of further long-term monitoring. The selection of the fenced plots was based on the abundance of the threatened target populations (Table 1). The size and site selection of a given enclosure also depended on its accessibility, the presence of natural features that supported the protection process (i.e., fencing), and homogeneity in its topography and vegetation.

**Table 1.** Characteristics of the 37 enclosures in the South Sinai mountainous region (Saint Katherine, Egypt).

| Enclosure No. | Location | Latitude (N) | Longitude (E) | Altitude (m) | Dominant Species | Habitat | Size (m²) |
|---|---|---|---|---|---|---|---|
| 1 | Wadi El-Arbain | 28°33′06.3″ | 33°56′59.4″ | 1570 | *Lycium shawii* | Slope | 7 |
| 2 | Wadi El-Arbain | 28°33′14.9″ | 33°56′56.8″ | 1580 | *Origanum syriacum* subsp. *sinaicum* | Slope | 74 |
| 3 | Wadi El-Arbain | 28°32′56.9″ | 33°56′56.2″ | 1650 | *Adiantum capillus-veneris* | Cave | 175 |
| 4 | Wadi El-Arbain | 28°32′41.0″ | 33°57′03.0″ | 1597 | *Tanacetum sinaicum* | Slope | 70 |
| 5 | Wadi El-Arbain | 28°32′14.7″ | 33°57′48.2″ | 1754 | *Juncus rigidus* | Bed | 73 |
| 6 | Wadi El-Arbain | 28°32′42.1″ | 33°57′48.5″ | 1739 | *Rosa arabica* | Terrace | 40 |
| 7 | Shagg Musa | 28°30′14.9″ | 33°56′56.8″ | 1880 | *Origanum syriacum* subsp. *sinaicum* | Gorge | 36 |
| 8 | Shagg Musa | 28°30′28.2″ | 33°58′10.7″ | 1920 | *Origanum syriacum* subsp. *sinaicum* | Slope | 90 |
| 9 | Shagg Musa | 28°31′26.3″ | 33°57′35.2″ | 1920 | *Phlomis aurea* | Slope | 84 |
| 10 | Shagg Musa | 28°31′30.8″ | 33°57′41.6″ | 1920 | *Artemisia herba-alba* | Slope | 21 |
| 11 | Shagg Musa | 28°31′28.2″ | 33°57′44.3″ | 2055 | *Cotoneaster orbicularis* | Slope | 24 |
| 12 | Shagg Musa | 28°31′25.2″ | 33°57′39.9″ | 2050 | *Crataegus x sinaica* | Gorge | 77 |
| 13 | Shagg Musa | 28°31′23.0″ | 33°57′36.6″ | 2092 | *Nepeta septemcrenata* | Slope | 51 |
| 14 | Wadi Graginya | 28°31′33.0″ | 33°58′12.0″ | 1920 | *Phlomis aurea* | Slope | 25 |
| 15 | Wadi Garginya | 28°31′31.0″ | 33°58′11.0″ | 1920 | *Phlomis aurea* | Slope | 25 |
| 16 | Wadi Garginya | 28°31′26.0″ | 33°58′09.0″ | 1920 | *Origanum syriacum* subsp. *sinaicum* | Cave | 17 |
| 17 | Musa's Gorge | 28°32′41.0″ | 33°58′54.0″ | 1590 | *Origanum syriacum* subsp. *sinaicum* | Gorge | 35 |
| 18 | Musa's Gorge | 28°32′38.0″ | 33°58′51.0″ | 1980 | *Atraphaxis spinosa* | Gorge | 25 |
| 19 | Musa's Gorge | 28°32′37.0″ | 33°58′50.0″ | 1770 | *Seriphidium herba-alba* | Gorge | 37 |
| 20 | Farsh El-Losa | 28°32′55.2″ | 33°58′17.2″ | 1996 | *Tanacetum sinaicum* | Bed | 68 |
| 21 | Farsh Shoeib | 28°33′07.6″ | 33°58′00.2″ | 1970 | *Seriphidium herba-alba* | Plain | 246 |
| 22 | Wadi El-Fara'a | 28°32′21.0″ | 33°57′56.1″ | 1860 | *Seriphidium herba-alba* | Slope | 135 |
| 23 | Wadi El-Fara'a | 28°32′23.4″ | 33°57′59.7″ | 1862 | *Artemisia judaica* | Slope | 25 |
| 24 | Wadi El-Fara'a | 28°32′29.2″ | 33°57′57.0″ | 1841 | *Pterocephalus sanctus* | Slope | 114 |
| 25 | Wadi El-Fara'a | 28°32′33.9″ | 33°57′54.0″ | 1843 | *Seriphidium herba-alba* | Slope | 34 |
| 26 | Wadi El-Fara'a | 28°32′48.2″ | 33°57′43.7″ | 1824 | *Seriphidium herba-alba* | Bed | 42 |
| 27 | Wadi El-Fara'a | 28°32′49.4″ | 33°57′41.2″ | 1862 | *Tanacetum sinaicum* | Bed | 6 |
| 28 | Gebel Katharina | 28°31′06.3″ | 33°57′20.9″ | 2377 | *Seriphidium herba-alba* | Foothill | 63 |
| 29 | Gebel Katharina | 28°31′07.7″ | 23°57′20.9″ | 2357 | *Seriphidium herba-alba* | Slope | 71 |
| 30 | Gebel El-Ahmar | 28°31′30.4″ | 33°56′58.3″ | 2161 | *Thymus decussatus* | Terrace | 50 |
| 31 | Wadi El-Dair | 28°33′28.0″ | 33°58′40.8″ | 1652 | *Echinops spinosus* | Bed | 40 |
| 32 | Wadi El-Dair | 28°33′31.3″ | 33°58′38.1″ | 1654 | *Echinops spinosus* | Gorge | 10 |
| 33 | Wadi El-Dair | 28°33′31.3″ | 33°58′26.4″ | 1586 | *Bituminaria bituminosa* | Bed | 62 |
| 34 | Wadi El-Dair | 28°33′37.3″ | 33°58′16.8″ | 1544 | *Artemisia judaica* | Bed | 77 |
| 35 | Gebel Muneiga | 28°32′48.0″ | 33°59′05.0″ | 1580 | *Seriphidium herba-alba* | Slope | 33 |
| 36 | Wadi Esbaiea | 28°33′59.8″ | 33°59′36.9″ | 1533 | *Alkanna orientalis* | Plain | 81 |
| 37 | Wadi Abu Tuweita | 28°34′39.8″ | 33°53′24.8″ | 1805 | *Salvia multicaulis* | Slope | 50 |

### 2.3. Vegetation Measurements

Within each enclosure, and a comparable area outside it, the plant species were recorded and their density (as individuals per 100 m²) and cover (%, as cm per 100 cm) were estimated applying the line-intercept method [13] using 2–5 lines along the length of each enclosure (depending on the enclosure size). The nomenclature used follows Boulos [14,15]. The height and diameter (based on 2–3 crown measurements) of all individuals of a certain species were measured to the nearest cm, and their means per species were then calculated. The biovolume was calculated as the mean of the height and diameter; this estimate is a better fit to the individuals' sizes than calculating their volume as a cylinder [16,17].

### 2.4. Data Analysis

The species richness within and outside the enclosures was calculated as the average number of species per enclosure in each case, and the species turnover (i.e., the extent of species replacement along environmental gradients: [18]) was calculated as the ratio between the total species in each case and its species richness. The relative evenness was calculated using the Shannon–Weaver index (H' = $-\Sigma\, p_i \log p_i$), whereas the relative concentration dominance was calculated using the Simpson index (D = $\Sigma\, p_i$), where $p_i$ is

the proportion of individuals in the *i*th species [19]. The curve of the species cove sequence was drawn as a logarithmic trending line using Microsoft Excel 2007 [20].

Detrended correspondence analysis (DCA), as a multivariate technique, was applied for the ordination of enclosures depending on the means of species cover within and outside them [21]. A simple linear correlation coefficient (*r*) was calculated to assess the significance of association between the DCA axes and species richness, plant cover, and altitude. An unpaired *t*-test was applied [22] to identify statistically significant differences in the density, cover, and biovolume of species inside and outside the enclosures. On the basis of the relative cover of species, the relative change increase or decrease (RID) in the richness, turnover, relative concentration of dominance, and absolute cover of species inside the enclosures compared with outside the enclosures was calculated as follows: RID = ((inside − outside)/outside) × 100.

## 3. Results

One hundred and two species were recorded in the mountainous ecosystem: 41 both within and outside the enclosures (Table 2), another 41 only within the enclosures, and 20 only outside the enclosures (Supplementary Materials). The density (individual per 100 m$^2$) and cover (%) of most species was higher within than outside. For many chamaephytic species, these differences were significant according to the *t*-test. Most of the 41 species recorded within the enclosures occupied only one to two enclosures with negligible cover.

**Table 2.** The mean densities and covers of the recorded species inside and outside the enclosures, with the number of enclosures in which the species were recorded. Values of *t* with $p \leq 0.05$ were written in bold.

| Species | Number of Enclosure | Density (Individual/100 m$^2$) | | | | Cover (%) | | | |
|---|---|---|---|---|---|---|---|---|---|
| | | Mean | | *t*-Value | *p* | Mean | | *t*-Value | *p* |
| | | Inside | Outside | | | Inside | Outside | | |
| *Achillea fragrantissima* | 11 | 6.2 | 7.0 | 1.48 | 0.17 | 4.2 | 10.0 | 1.24 | 0.26 |
| *Alkanna orientalis* | 24 | 15.6 | 11.4 | 2.02 | 0.06 | 9.0 | 6.0 | 0.76 | 0.46 |
| *Anarrhinum pubescens* | 4 | 25.5 | 9.5 | 1.06 | 0.36 | 6.6 | 3.9 | 0.81 | 0.50 |
| *Andrachne aspera* | 5 | 50.3 | 17.2 | 1.18 | 0.30 | 3.7 | 4.2 | 0.22 | 0.85 |
| *Seriphidium herba-alba* | 25 | 52.3 | 33.5 | **3.59** | **0.00** | 16.1 | 3.0 | **4.73** | **0.00** |
| *Artemisia judaica* | 12 | 81.1 | 50.4 | 1.51 | 0.16 | 23.5 | 15.7 | 0.94 | 0.39 |
| *Astragalus asterias* | 5 | 32.8 | 8.8 | 1.58 | 0.19 | 2.7 | 0.9 | **4.22** | **0.00** |
| *Atraphaxis spinosa* | 5 | 14.3 | 11.8 | 0.43 | 0.69 | 6.9 | 4.4 | 1.68 | 0.19 |
| *Ballota undulata* | 19 | 10.7 | 5.0 | **2.32** | **0.03** | 4.0 | 1.8 | 1.55 | 0.14 |
| *Bituminaria bituminosa* | 4 | 44.0 | 9.5 | 1.53 | 0.22 | 6.7 | 3.9 | 1.29 | 0.33 |
| *Chiliadenus montanus* | 16 | 23.3 | 5.3 | 1.71 | 0.11 | 11.4 | 2.1 | **2.64** | **0.03** |
| *Crataegus x sinaica* | 8 | 5.0 | 7.0 | 2.26 | 0.06 | 13.4 | 3.3 | **3.37** | **0.02** |
| *Dianthus sinaicus* | 9 | 13.0 | 11.8 | 1.99 | 0.08 | 1.3 | 2.0 | 0.52 | 0.70 |
| *Echinops macrochaetus* | 4 | 19.3 | 9.0 | **7.17** | **0.00** | 4.3 | 18.3 | 0.43 | 0.71 |
| *Echinops spinosus* | 22 | 24.2 | 24.2 | 1.16 | 0.26 | 8.7 | 6.0 | **2.08** | **0.05** |
| *Equisetum ramosissimum* | 1 | 178.0 | 12.0 | 0.00 | 1.00 | 23.7 | 2.2 | 0.00 | 1.00 |
| *Fagonia arabica* | 5 | 3.0 | 10.5 | 0.94 | 0.40 | 4.5 | 2.0 | 0.59 | 0.60 |
| *Fagonia mollis* | 8 | 29.8 | 13.3 | 1.62 | 0.15 | 7.3 | 6.7 | 0.09 | 0.93 |
| *Globularia arabica* | 1 | 10.0 | 4.0 | 0.00 | 1.00 | 8.2 | 3.6 | 0.00 | 1.00 |
| *Juncus acutus* | 1 | 21.0 | 16.0 | 0.00 | 1.00 | 29.2 | 8.6 | 0.00 | 1.00 |
| *Juncus rigidus* | 7 | 13.4 | 18.3 | 0.63 | 0.55 | 13.1 | 11.8 | 0.59 | 0.58 |
| *Launaea nudicaulis* | 6 | 2.8 | 7.0 | 0.59 | 0.58 | 1.7 | 1.0 | 0.10 | 0.94 |
| *Lavandula stricta* | 1 | 32.0 | 21.0 | 0.00 | 1.00 | 16.2 | 8.8 | 0.00 | 1.00 |
| *Matthiola arabica* | 8 | 9.0 | 4.4 | 0.73 | 0.48 | 3.0 | 0.8 | 1.27 | 0.26 |
| *Mentha longifolia* | 5 | 33.7 | 7.5 | 0.92 | 0.41 | 3.9 | 9.2 | 0.17 | 0.87 |
| *Minuartia meyeri* | 3 | 3.3 | 9.0 | 0.18 | 0.87 | 6.5 | 0.5 | 0.86 | 0.55 |
| *Nepeta septemcrenata* | 16 | 7.2 | 9.4 | 0.99 | 0.34 | 8.4 | 3.3 | 1.70 | 0.13 |
| *Origanum syriacum sinaicum* | 9 | 19.7 | 7.0 | 1.96 | 0.09 | 20.6 | 11.6 | **3.83** | **0.00** |

**Table 2.** *Cont.*

| Species | Number of Enclosure | Density (Individual/100 m²) | | | | Cover (%) | | | |
|---|---|---|---|---|---|---|---|---|---|
| | | Mean | | *t*-Value | *p* | Mean | | *t*-Value | *p* |
| | | Inside | Outside | | | Inside | Outside | | |
| *Phlomis aurea* | 26 | 16.1 | 9.3 | **2.09** | **0.05** | 16.8 | 12.7 | 1.78 | 0.09 |
| *Plantago sinaica* | 11 | 19.9 | 16.8 | 1.67 | 0.13 | 6.1 | 1.0 | 1.32 | 0.24 |
| *Pterocephalus sanctus* | 10 | 14.7 | 11.0 | 1.11 | 0.29 | 14.9 | 2.8 | 1.06 | 0.33 |
| *Reseda arabica* | 1 | 5.0 | 8.0 | 0.00 | 1.00 | 6.7 | 5.0 | 0.00 | 1.00 |
| *Spergularia diandra* | 11 | 10.8 | 21.3 | 1.05 | 0.32 | 0.6 | 0.4 | 0.66 | 0.54 |
| *Stachys aegyptiaca* | 21 | 17.0 | 19.1 | 0.12 | 0.91 | 10.9 | 4.5 | **3.21** | **0.00** |
| *Stipagrostis ciliata* | 3 | 16.5 | 6.7 | 0.55 | 0.64 | 6.0 | 3.1 | 0.84 | 0.49 |
| *Tanacetum sinaicum* | 31 | 17.5 | 17.4 | **2.28** | **0.03** | 12.3 | 8.9 | 1.63 | 0.12 |
| *Teucrium polium* | 25 | 16.7 | 17.2 | 0.32 | 0.75 | 6.4 | 3.5 | 1.87 | 0.08 |
| *Thymus decussatus* | 6 | 33.0 | 16.0 | 2.42 | 0.06 | 11.6 | 4.9 | 1.91 | 0.12 |
| *Verbascum sinaiticum* | 8 | 4.7 | 2.5 | 2.01 | 0.09 | 2.7 | 2.3 | 0.46 | 0.67 |
| *Zilla spinosa* | 9 | 12.4 | 6.0 | 1.08 | 0.31 | 5.6 | 0.1 | 1.59 | 0.17 |
| *Zilla spinosa* subsp. *spinosa* | 4 | 6.5 | 6.0 | 0.27 | 0.81 | 2.6 | 4.6 | 0.08 | 0.94 |

Some species, such as *Adantium capillus-vernis*, *Primula boveana*, and *Veronica angalis-aquatica*, inhabited the moist ground inside caves and streams. In addition, many of the species outside the enclosures had negligible cover and minor occurrence (only one enclosure), and some were alien (e.g., *Althaea liudwigii*) or weed species (e.g., *Malva parviflora*). The most prominent species occurring both within and outside the enclosures were chamaephytes (58.5%), which had been over-grazed (65.9%) and over-collected for medicinal use (48.8%) (Tables 3 and 4). The plants that were negatively affected by heavy grazing were *Zilla spinosa*, *Tanacetum sinaicum*, *Seriphidium herba-alba*, and *Silene schimperiana*.

**Table 3.** Biological spectrum of the recorded species both inside and outside (inside-outside), only inside, and only outside the protected enclosures.

| Life Form | Inside-Outside | | Inside | | Outside | |
|---|---|---|---|---|---|---|
| | Actual Number | % of Total Species | Actual Number | % of Total Species | Actual Number | % of Total Species |
| Therophytes | 4 | 9.6 | 11 | 26.8 | 7 | 35.0 |
| Chamaephytes | 24 | 58.5 | 12 | 29.3 | 8 | 40.0 |
| Cryptophytes | 1 | 2.4 | 7 | 17.1 | | |
| Phanerophytes | | | 4 | 9.8 | 2 | 10.0 |
| Hemicryptophytes | 12 | 29.3 | 7 | 17.1 | 3 | 15.0 |
| Total species | 41 | | 41 | | 20 | |

**Table 4.** Common services offered by the species inside-outside, only inside, and only outside the protected enclosures.

| Life Form | Inside-Outside | | Inside | | Outside | |
|---|---|---|---|---|---|---|
| | Actual Number | % of Total Species | Actual Number | % of Total Species | Actual Number | % of Total Species |
| Grazing | 27 | 65.9 | 23 | 56.1 | 9 | 45.0 |
| Medicinal | 20 | 48.8 | 10 | 24.4 | 10 | 50.0 |
| Fuel | 6 | 14.6 | 6 | 14.6 | 3 | 15.0 |
| Edible | 4 | 4.1 | 3 | 7.3 | 3 | 15.0 |
| Total species | 41 | | 41 | | 20 | |

The total cover was greater within (3.7%) compared with outside the enclosures (2.1%). In addition, the vegetation within the enclosures included a higher number of species, species richness, relative evenness, and total cover but lower species turnover and relative concentration of dominance (Table 5). The species abundance curve of the vegetation inside the enclosures deviated toward the MacArthur species distribution, whereas the vegetation outside the enclosures deviated towards the geometric species distribution (Figure 2). After applying the *t*-test, the biovolumes of 13 species (most of which were chamaephytes) had significantly higher values inside than outside, with *Cratagus x sinaica* and *Flomis aurea* having the highest values (Figure 3).

**Table 5.** Comparison between the total cover of species as well as the species diversity indices inside and outside the enclosures. Relative change increase or decrease (RID) = (inside − outside)/outside.

| Diversity Component | Inside | Outside | RID |
|---|---|---|---|
| Total cover (m/100 m) | 3.7 | 2.1 | 0.78 |
| Total species | 82.0 | 61.0 | 0.34 |
| Species richness ($\bar{\alpha}$) | 12.60 | 8.40 | 0.50 |
| Relative evenness (H) | 1.50 | 1.30 | 0.15 |
| Relative concentration of dominance (C) | 0.05 | 0.08 | −0.38 |
| Species turnover ($\beta w$) | 5.50 | 6.30 | −0.13 |

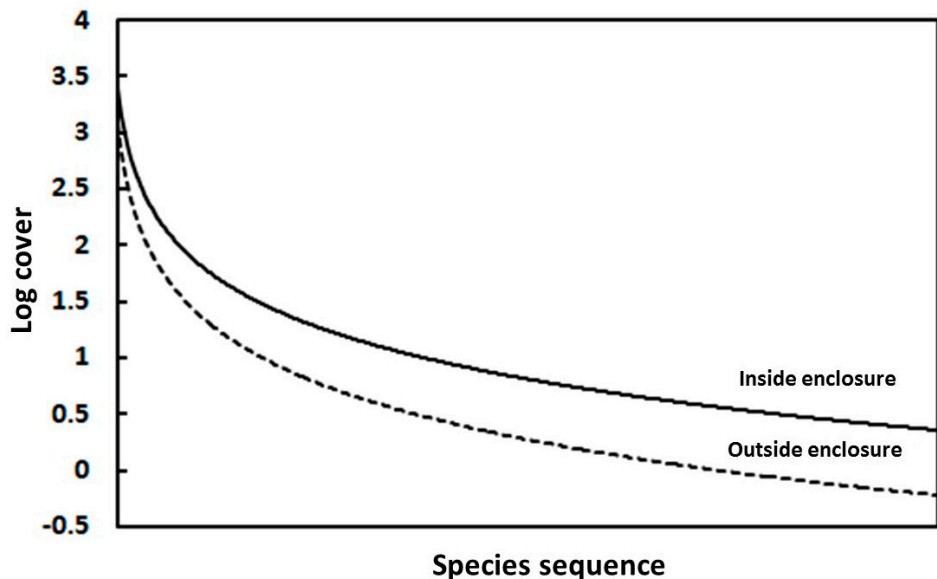

**Figure 2.** Logarithmic trend line of the species cover sequence inside and outside the enclosures.

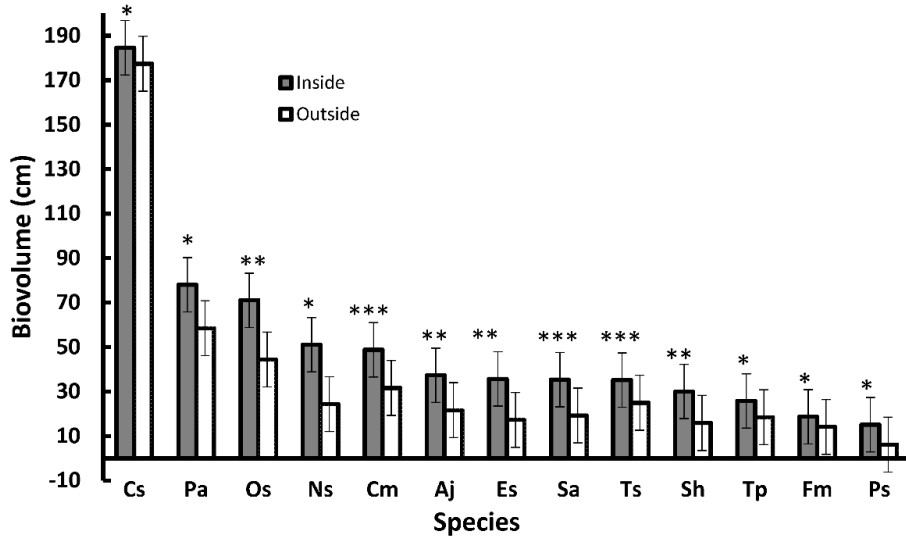

**Figure 3.** The mean biovolume of the species with significant higher values inside compared with outside the enclosures. Sh: *Seriphidium herba-alba*, Aj: *Artemisia judaica*, Cm: *Chiliadenus montanus*, Cs: *Crataegus x sinaica*, Es: *Echinops spinosus*, Fm: *Fagonia mollis*, Ns: *Nepeta septemcrenata*, Os: *Origanum syriacum* subsp. *sinaicum*, Pa: *Phlomis aurea*, Ps: *Plantago sinaica*, Sa: *Stachys aegyptiaca*, Ts: *Tanacetum sinaicum*, and Tp: *Teucrium polium*. Vertical bars are the standard errors of the means. *: $p \leq 0.05$, **: $p \leq 0.01$, and ***: $p \leq 0.001$ according to the *t*-test.

Applying the DCA ordination, the first axis was significantly correlated with species richness ($r = 0.88$, $p < 0.001$), total cover ($r = 0.78$, $p < 0.001$), and altitude ($r = 0.55$, $p < 0.001$); the second axis was significantly correlated with species richness ($r = 0.65$, $p < 0.001$) and altitude ($r = 0.51$, $p < 0.001$). Nineteen enclosures had progressive cover along the first axis in addition to species richness and altitude along both axes, whereas the reverse was true regarding 12 enclosures (Figure 4). On the other hand, no vegetation was recorded outside six enclosures.

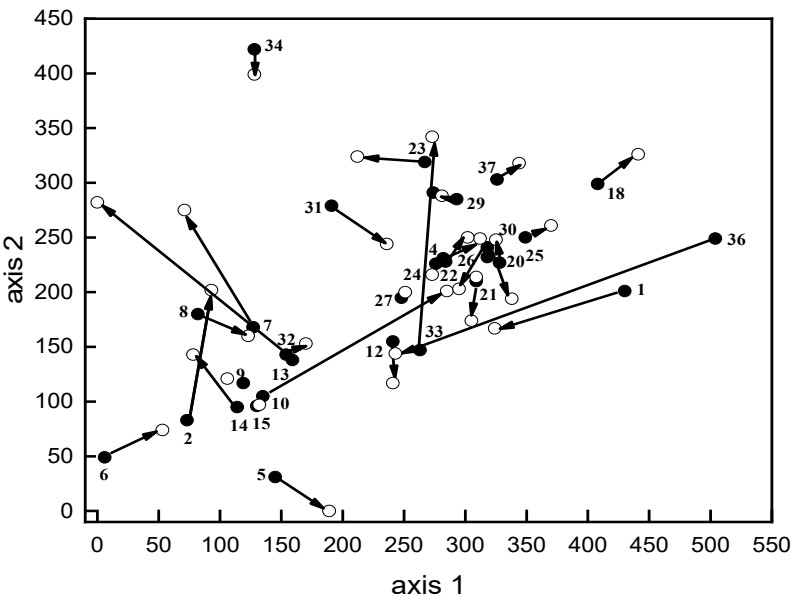

**Figure 4.** The detrended correspondence analysis (DCA) ordination of the enclosures according to the importance value data (cover). Open circles represent the inside vegetation, solid circles represent the outside vegetation, and arrows represent the direction of change.

## 4. Discussion

After six years of protection, the vegetation change in the mountainous region in South Sinai was examined using the abundance (density), dominance (cover), and size dimension (biovolume) data. The detrended correspondence analysis of a combined set of data within and outside the enclosures indicated that the overall changes in vegetation composition were relatively small. Many enclosures shifted to a progressive position, in terms of the species richness, altitude, and plant cover, along the ordination axes from the outside to the inside. It is likely that the cessation of over-grazing and other human impacts increased the diversity inside the enclosures. Numerous common species appear to be of high palatability as food and, thus, are rare outside (e.g., *Ballota damascene*, *Helianthemum sancti-antonii*, and *Primula boveana*). After, these species regenerated in various habitats, and there were more of them [23].

Inside the enclosures, the protection process led to an improvement in the vegetation and its individual populations, with respect to the plant cover, diversity indices, and biovolume. Many of these plant populations are either endemic or threatened species of high conservational value. In some enclosures, the improvement of vegetation was associated with the deterioration of certain target species, and in other enclosures the total vegetation had deteriorated. To explain the reasons of such deterioration, further studies, particularly at the population level, are recommended to be performed.

Improvement in the vegetation due to full or partial protection has also been investigated by Kassas [24] at Ras El-Hikma (Egypt) and Ayyad [25], and Ayyad and El-Kadi [26] in the Western Mediterranean Desert of Egypt. The study of Teketay et al., [3] on the woodland in northern Botswana, recorded that the enclosure had a seven-times higher mean density of woody species compared with outside the enclosure, with exceptional

regeneration of seedlings inside compared with outside. In addition, Gebremedihin et al.'s study, [4] in the highlands of Tigray (Northern Ethiopia), reported that the restoration of degraded drylands through enclosures enhanced the woody species diversity, soil nutrient diversity, and species richness inside the enclosures compared with the grazing lands.

Conversely, previous related studies (e.g., [27]) reported that full protection over an extended period against over-grazing and other human activities was not advantageous to the natural vegetation. Partial protection via rotation-grazing should be tested as an efficient conservational tool. The findings of Belgacem et al., [28] regarding the effect of livestock grazing on plant cover and species diversity in the Qatar desert rangelands, revealed considerable positive effects of protection on the vegetation parameters. However, their results emphasized a negative effect of the long-term protection. Short-term protection followed by light grazing was found to be more sustainable than long-term protection, in terms of the plant cover, species richness, and biovolume. Some species were aliens, such as *Althaea ludwigii*, which was recorded outside two enclosures (causal species), or weed species, such as *Malva parviflora*, which was recorded in one enclosure only [29].

Limiting grazing was found to be a more effective factor than water in conserving desert plants, particularly edible plants. In the present study, most of the species with significantly higher density and cover within compared to outside the enclosures were chamaephytes (i.e., shrubs). In the Junggar Basin (China), Rong et al. [2] reported that excluding sheep grazing from a desert steppe for eight years increased the plant cover and approximately tripled the biomass of the standing vegetation, particularly the shrub component. As reported by Tarhouni et al., [30] in relation to the southern dry deserts of Tunisia, these life forms encompass plants that are highly sensitive to human and animal disturbances. This finding is comparable to the study of El-Keblawy [31] in the UAE. Perennating buds near the ground are usually more vulnerable to destruction by grazing animals [31].

Protection against over-exploitation could provide an opportunity for the regeneration of vegetation and the improvement of protection. This study suggests an overall phytomass increase due to protection for six years (as reflected by the total plant cover). A similar conclusion was made by Pearson [32] in his study on the primary production in grazed and ungrazed desert communities of Idaho, USA. He reported an increase in the standing crop biomass of 45% following 11 years of protection. In addition, Shaltout and El-Ghareeb [27] reported an increase of 36% in the above-ground biomass associated with a decrease of 10% in the below-ground biomass after four years of protection of the non-saline depressions in the Western Mediterranean Desert of Egypt.

The protection of vegetation leads to an initial increase in density and standing crop phytomass. The stress created by the proximity of neighbors may be absorbed in an increased mortality risk for whole plants or their parts, reduced reproductive output, reduced growth rate, and delay of maturity and reproduction [33]. Researchers hypothesized that the plant litter accumulation in protected or lightly grazed stands apparently prevents gap creation and, hence, seedling establishment [34]. This may partially explain why the quantity of standing dead material was greater within, compared to outside, many protected sites, as the standing dead shoots are typically removed by grazing animals [32].

The domestic grazing animals in the SKP are mainly goats, camels, and feral donkeys, which use the wadis with differing regularity [35]. For example, locations close to the Saint Katherine and Bedouin communities contained the highest quantities of goat dung (personal observations), indicating that goats intensely graze the majority of these sites. In addition, large numbers of camels are present in Wadi El-Arbain, which is easily accessible and heavily used by tourists and camels, and feral donkeys are concentrated in locations with high plant cover [35].

## 5. Conclusions

Using enclosures for the protection of vegetation against over-grazing and over-cutting in the South Sinai mountainous region, for a period of six years, resulted in the

improvement of its vegetation in terms of the total density, total cover, and species diversity. Such variation in vegetation during the study period might be attributable to factors that were not controlled by authors, for instance, climatic factors (no in situ climatic data exist). In mountainous regions similar to the area of the present study, which are characterized by topographic and physiographic heterogeneity, the variations in the microclimate play a major role in governing the natural vegetation. In conclusion, the enclosures are instrumental in improving the species cover and species diversity; thus, enclosures have the potential to contribute to the resilience of vegetation in arid lands.

**Supplementary Materials:** The following are available online at https://www.mdpi.com/1424 -2818/13/3/113/s1, Table S1. Life forms and common services offered by the plant species in the Saint Katherine (South Sinai, Egypt). Life forms are: PH: phanerophytes, CH: chamaephytes, CR: cryptophytes, HC: hemicryptophytes, and TH: therophytes. Common services are: GR: grazing, ME: medicine, FU: fuel, and ED: edible. Table S2. Density and cover of species found only inside the enclosures, with the number of enclosures in which the species were recorded. Table S3. Density and cover of species found only outside the enclosures, with the number of enclosures in which the species were recorded.

**Author Contributions:** Conceptualization, K.H.S.; methodology, K.H.S., E.M.E., Y.M.A.-S., and S.Z.H.; software, Y.M.A.-S., S.K.S., and S.A.E.-M.; formal analysis, K.H.S., Y.M.A.-S., and S.Z.H.; investigation, K.H.S.; resources, E.M.E.; data curation, K.H.S. and Y.M.A.-S.; Writing—Original draft preparation, K.H.S.; Writing—Review and editing, E.M.E., Y.M.A.-S., S.Z.H., S.K.S., and S.A.E.-M.; visualization, E.M.E.; supervision, K.H.S.; project administration, E.M.E.; funding acquisition, E.M.E. All authors have read and agreed to the published version of the manuscript.

**Funding:** This research was funded by the Scientific Research Deanship at King Khalid University and the Ministry of Education in Saudi Arabia through the project number IFP-KKU-2020/3.

**Institutional Review Board Statement:** Not applicable.

**Informed Consent Statement:** Not applicable.

**Data Availability Statement:** Data is contained within the article and Supplementary Material.

**Conflicts of Interest:** The authors declare no conflict of interest.

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
