# Peer review of "Effect of Protection of Mountainous Vegetation against Over-Grazing and Over-Cutting in South Sinai, Egypt"

_diversity, doi:10.3390/d13030113_

Round 1
Reviewer 1 Report
Dear Authors,
The submitted manuscript is of good quality, deserving publication after its revision. My main comments are below.
The title and abstract are well written, except for needs to make some corrections in English. Moreover (and it concerns the whole manuscript), if the study has been carried out in 2021-2018, then the study period is seven (1 - 2012, 2 - 2013, 3 - 2014, 4 - 2015, 5 -2016, 6 - 2017, 7 - 2018) years, but not "six-year period", as it is stated through the manuscript.
In the list of key words, I suggest to exclude words and sentences represented in the title of the manuscript. Please, make the revision.
The section Introduction needs to be revised, too. Now, Introduction is focused on South Sinai only (even not on the entire Egypt). As the Diversity is a high-quality journal with a broad international audience, I strongly recommend to add one more (first in the section) paragraph devoted to a brief overview of literature devoted to the research topic, i.e. the use of small-area sites to protect biodiversity. By the way, I have recently found some studies, which are highly similar to this manuscript, namely http://dx.doi.org/10.24189/ncr.2018.001 (this study [in my view] has especially similar general results, obtained in Ethiopia), http://doi.org/10.1186/s13717-018-0116-x, http://doi.org/10.1002/ldr.2420, http://doi.org/10.1111/grs.12048, and others. The main aim for this is to highlight the international relevance of studies devoted to the use of enclosure (exclosure) for nature restoration.
After revision, the section Material and Methods looks relatively well. But it still needs to be revised. For example, in Fig.1, the point of studied sites should not be marked by yellow because in this way, they are poorly visible. I suggest to use red or black color to indicate the study sites in better way.
And one more important suggestion concerns terminology. In the manuscript, authors use the term "vegetation", but they didn't study plant communities. Please, note that if you use plant communities, you may say "vegetation" (i.e. a set of plant communities), but if you study plant species, it is "flora" (i.e. species composition or list of species).
In present form, the section Results became better, I think. Here minor corrections are needed only. For example, in captions of all tables and figures, please, add designations (explanation) of all abbreviations used in these tables and figures.
However, my main suggestions is here: I suggest to enlarge descriptions of figures and tables presented here. Now, this description is very short, with a lack of details. of course, interpretation and discussion of the obtained results will be present in the section Discussion, but the description should be enlarged, too.
The section Discussion should be revised, too. So, the first paragraph is rather description of results. However, the rest part of the section is poorly connected with the section Results. The results, presented in each table and each figure, must be discussed in the section Discussion, with the comparison of them with data published previously. And here (as well as in Introduction, too) I strongly recommend to not be limited by data from South Sinai or Egypt only, but, please, involve data from other regions in Africa and (maybe) outside it.
The section Conclusions is too brief and contains irrelevant materials. For example, the sentence "This improvement was evaluated based on the comparison between the total cover, total species and species richness within and outside the enclosures" is rather Results. here, please, present the main implications / conclusions reflected the results and their discussion. Now there is a poor connection between Conclusions and obtained results. Please, improve this section with a more focus on your results.
For the whole manuscript: English quality needs to be improved.
Author Response
27 February 2021
Prof. Dr. Michael Wink
Editor-in-Chief
Diversity
Dear Prof. Wink,
Please find attached the revised manuscript titled ‘Effect of protection of mountainous vegetation against over-grazing and over-cutting in South Sinai, Egypt’. Manuscript ID (diversity-1128777), authored by Kamal H. Shaltout, Ebrahem M. Eid, Yassin M. Al-Sodany, Selim Z. Heneidy, Salma K. Shaltout, and Safaa A. El-Masry.
On behalf of my co-authors, I thank you very much for giving us the opportunity to revise our manuscript. We appreciate the positive and constructive comments and suggestions provided by the reviewers on our manuscript. We have carefully studied the reviewers’ comments and have made revisions that are highlighted in yellow in the revised version of the manuscript. We have tried our best to revise our manuscript according to the reviewers’ comments. Please find attached the revised version of our manuscript, which we would like to submit for your kind consideration. Once again, we would like to express our great appreciation to you and the reviewers for the comments on our manuscript.
Please find below our detailed responses to each of the points raised.
---------------------------------------------------------------------------------------------------------------------
Reviewers' comments:
Reviewer #1
- The title and abstract are well written, except for needs to make some corrections in English. Moreover (and it concerns the whole manuscript), if the study has been carried out in 2012-2018, then the study period is seven (1 - 2012, 2 - 2013, 3 - 2014, 4 - 2015, 5 -2016, 6 - 2017, 7 - 2018) years, but not "six-year period", as it is stated through the manuscript
Response: First of all, thanks so much Sir for your positive and constructive comments and suggestions. To meet the standards of the journal, an English Language Editing Service provided by MDPI English Editing Services (Project no. english-27330) was used. Please see the attached certificate of language editing. The present study extended from March 2012 to March 2018 (a total of six years). We indicated this in the text.
---------------------------------------------------------------------------------------------------------------------
- In the list of key words, I suggest to exclude words and sentences represented in the title of the manuscript. Please, make the revision.
Response: We omitted 2 words (overgrazing and overcutting) and added a new one (Saint Katherine).
---------------------------------------------------------------------------------------------------------------------
- The section Introduction needs to be revised, too. Now, Introduction is focused on South Sinai only (even not on the entire Egypt). As the Diversity is a high-quality journal with a broad international audience, I strongly recommend to add one more (first in the section) paragraph devoted to a brief overview of literature devoted to the research topic, i.e. the use of small-area sites to protect biodiversity. By the way, I have recently found some studies, which are highly similar to this manuscript, namely http://dx.doi.org/10.24189/ncr.2018.001 (this study [in my view] has especially similar general results, obtained in Ethiopia), http://doi.org/10.1186/s13717-018-0116-x, http://doi.org/10.1002/ldr.2420, http://doi.org/10.1111/grs.12048, and others. The main aim for this is to highlight the international relevance of studies devoted to the use of enclosure (enclosure) for nature restoration.
Response: Thanks for giving us the links of these 4 interesting publications. We used them for supporting the Introduction and Discussion. Also, we added them to the list of references.
---------------------------------------------------------------------------------------------------------------------
- After revision, the section Material and Methods looks relatively well. But it still needs to be revised.
- For example, in Fig.1, the point of studied sites should not be marked by yellow because in this way, they are poorly visible. I suggest to use red or black color to indicate the study sites in better way.
Response: We tried red and black color and there was no contrast. Thus, the point of studied sites was marked by blue color.
---------------------------------------------------------------------------------------------------------------------
- One more important suggestion concerns terminology: in the manuscript, authors use the term "vegetation", but they didn't study plant communities. Please, note that if you use plant communities, you may say "vegetation" (i.e. a set of plant communities), but if you study plant species, it is "flora" (i.e. species composition or list of species).
Response: Of course, we know the difference between the 2 term. Here we used the total cover and diversity indices that are considered as macro-community characters.
---------------------------------------------------------------------------------------------------------------------
- In present form, the section Results became better, I think. Here minor corrections are needed only.
- For example, in captions of all tables and figures, please, add designations (explanation) of all abbreviations used in these tables and figures.
Response: We revised this in all the captions.
---------------------------------------------------------------------------------------------------------------------
However, my main suggestions here are: I suggest to enlarge descriptions of figures and tables presented here. Now, this description is very short, with a lack of details. of course, interpretation and discussion of the obtained results will be present in the section Discussion, but the description should be enlarged, too.
Response: We carried out some minor modifications to enlarge descriptions of figures and tables.
---------------------------------------------------------------------------------------------------------------------
- The section Discussion should be revised, too. So, the first paragraph is rather description of results. However, the rest part of the section is poorly connected with the section Results. The results, presented in each table and each figure, must be discussed in the section Discussion, with the comparison of them with data published previously. Here (as well as in Introduction, too), I strongly recommend to not be limited by data from South Sinai or Egypt only, but, please, involve data from other regions in Africa and (maybe) outside it.
Response: We used the references you had sent to improve the discussion.
---------------------------------------------------------------------------------------------------------------------
- The section Conclusions is too brief and contains irrelevant materials. For example, the sentence "This improvement was evaluated based on the comparison between the total cover, total species and species richness within and outside the enclosures" is rather Results. here, please, present the main implications / conclusions reflected the results and their discussion. Now there is a poor connection between Conclusions and obtained results. Please, improve this section with a more focus on your results.
Response: Also the conclusions were subjected to minor omitting and addition for supporting it.
---------------------------------------------------------------------------------------------------------------------
- For the whole manuscript: English quality needs to be improved.
Response: To meet the standards of the journal, an English Language Editing Service provided by MDPI English Editing Services (Project no. english-27330) was used. Please see the attached certificate of language editing.
---------------------------------------------------------------------------------------------------------------------
I hope the explanation given above adequately addresses all the reviewers’ comments. I would appreciate if the revised version of our manuscript would be considered for publication in Diversity.
Sincerely,
Ebrahem M. Eid
[Kafrelsheikh University]
[Botany Department, Faculty of Science, Kafrelsheikh University, Kafr El-Sheikh 33516, Egypt]
[Phone number: 002010 22648840]
[Email address: ebrahem.eid@sci.kfs.edu.eg]

Reviewer 2 Report
Dear authors
I appreciated your justifications, compliance with remarks, mine and those of reviewer #1's including the ones you did not agree with. I think the text is clearer now. Nevertheless, I have a few final editing suggestions:
28 – the vegetation pattern inside the enclosure became more or less stable, presumably due to (…)
29 ‘which led also to increase in’
88-91 I understand the justification for the difference in topography, but you have to explain the criteria to find their area. Add, per example, ‘the area of fenced plots was estimated as a proportion of a priori diversity of each site’
English language is still problematic. Please, have the article revised by a English native speaker.
Best wishes.
reviewer #2
Author Response
27 February 2021
Prof. Dr. Michael Wink
Editor-in-Chief
Diversity
Dear Prof. Wink,
Please find attached the revised manuscript titled ‘Effect of protection of mountainous vegetation against over-grazing and over-cutting in South Sinai, Egypt’. Manuscript ID (diversity-1128777), authored by Kamal H. Shaltout, Ebrahem M. Eid, Yassin M. Al-Sodany, Selim Z. Heneidy, Salma K. Shaltout, and Safaa A. El-Masry.
On behalf of my co-authors, I thank you very much for giving us the opportunity to revise our manuscript. We appreciate the positive and constructive comments and suggestions provided by the reviewers on our manuscript. We have carefully studied the reviewers’ comments and have made revisions that are highlighted in yellow in the revised version of the manuscript. We have tried our best to revise our manuscript according to the reviewers’ comments. Please find attached the revised version of our manuscript, which we would like to submit for your kind consideration. Once again, we would like to express our great appreciation to you and the reviewers for the comments on our manuscript.
Please find below our detailed responses to each of the points raised.
---------------------------------------------------------------------------------------------------------------------
Reviewers' comments:
Reviewer #2
I appreciated your justifications, compliance with remarks, mine and those of reviewer #1's including the ones you did not agree with. I think the text is clearer now. Nevertheless, I have a few final editing suggestions:
- 28: the vegetation pattern inside the enclosure became more or less stable, presumably due to (…)
Response: First of all, thanks so much Sir for your positive and constructive comments and suggestions. We did this correction.
---------------------------------------------------------------------------------------------------------------------
- 29: ‘which led also to increase in’
Response: Also we corrected this.
---------------------------------------------------------------------------------------------------------------------
- 88-91: I understand the justification for the difference in topography, but you have to explain the criteria to find their area. Add, per example, ‘the area of fenced plots was estimated as a proportion of a priori diversity of each site’
Response: We added this statement after minor modification.
---------------------------------------------------------------------------------------------------------------------
- English language is still problematic. Please, have the article revised by an English native speaker.
Response: To meet the standards of the journal, an English Language Editing Service provided by MDPI English Editing Services (Project no. english-27330) was used. Please see the attached certificate of language editing.
---------------------------------------------------------------------------------------------------------------------
I hope the explanation given above adequately addresses all the reviewers’ comments. I would appreciate if the revised version of our manuscript would be considered for publication in Diversity.
Sincerely,
Ebrahem M. Eid
[Kafrelsheikh University]
[Botany Department, Faculty of Science, Kafrelsheikh University, Kafr El-Sheikh 33516, Egypt]
[Phone number: 002010 22648840]
[Email address: ebrahem.eid@sci.kfs.edu.eg]

Reviewer 3 Report
The revised manuscript is improved and clearer in the presentation of the hypotheses, the experimental design and in the discussion of the results obtained.
Author Response
27 February 2021
Prof. Dr. Michael Wink
Editor-in-Chief
Diversity
Dear Prof. Wink,
Please find attached the revised manuscript titled ‘Effect of protection of mountainous vegetation against over-grazing and over-cutting in South Sinai, Egypt’. Manuscript ID (diversity-1128777), authored by Kamal H. Shaltout, Ebrahem M. Eid, Yassin M. Al-Sodany, Selim Z. Heneidy, Salma K. Shaltout, and Safaa A. El-Masry.
On behalf of my co-authors, I thank you very much for giving us the opportunity to revise our manuscript. We appreciate the positive and constructive comments and suggestions provided by the reviewers on our manuscript. We have carefully studied the reviewers’ comments and have made revisions that are highlighted in yellow in the revised version of the manuscript. We have tried our best to revise our manuscript according to the reviewers’ comments. Please find attached the revised version of our manuscript, which we would like to submit for your kind consideration. Once again, we would like to express our great appreciation to you and the reviewers for the comments on our manuscript.
Please find below our detailed responses to each of the points raised.
---------------------------------------------------------------------------------------------------------------------
Reviewers' comments:
Reviewer #3
The revised manuscript is improved and clearer in the presentation of the hypotheses, the experimental design and in the discussion of the results obtained.
Response: Thanks so much Sir for your positive and constructive comments and suggestions.
---------------------------------------------------------------------------------------------------------------------
I would appreciate if the revised version of our manuscript would be considered for publication in Diversity.
Sincerely,
Ebrahem M. Eid
[Kafrelsheikh University]
[Botany Department, Faculty of Science, Kafrelsheikh University, Kafr El-Sheikh 33516, Egypt]
[Phone number: 002010 22648840]
[Email address: ebrahem.eid@sci.kfs.edu.eg]

Round 2
Reviewer 1 Report
Dear Authors,
I would like to express my congratulations to you for taking into account of all of my previous comments. I found that you have improved abstract and the list of key words.
In Introduction, you used the references suggested by me. Although I hoped that some else publications could be involved in consideration, I think that the used references are well involved in both Introduction and Discussion.
The titles of figures and tables became self-sustainable, and they look well.
I am especially grateful to you for re-writing the section Conclusions, which is more connected with the results and their discussion, now. By the way, the Discussion is appropriately supplemented during the revision.
Thus, I am glad to recommend this manuscript for publication in the journal Diversity.
This manuscript is a resubmission of an earlier submission. The following is a list of the peer review reports and author responses from that submission.
Round 1
Reviewer 1 Report
The present manuscript could be published in the journal Diversity after its revision according to the raised questions.
The title doesn't reflect the content of the manuscript, as authors didn't analyze any anthropogenic factors, as well as "protection" sounds too generally because authors didn't study all aspects of the nature protection apart of the enclosures (e.g. no national parks, nature reserves, etc.).
Abstract needs in revision, too. So, the study period duration is not indicated. Then, it is wrong that authors say about temporal changes (e.g. "After fencing, all species regenerated in various enclosures...") because they didn't investigate vegetation patterns at the same sites before and after fencing, but they investigated vegetation different plots at the same time. Therefore, authors must re-write appropriately the abstract and the whole manuscript on the basis of results really obtained in the study, i.e. comparison of vegetation patterns inside and outside of enclosures.
The list of key words look to be correct.
The whole manuscript doesn't have appropriate contents due to the graphical contents covers about 65-70% (11 out of 16 pages) of the main text (from Introduction to Conclusions, inclusively), while it is widely accepted that tables and figures should not cover more than 30-40% of the main text in articles published in journals.
Concerning the sections Introduction and Discussion are of regional scale in terms of the used references.
I should note that authors should make sure that they cite the most appropriate references, i.e., either those that discovered the respective problem of international level or those that are of the highest quality or those, which are the newest. In this paper, authors used most of the references, for which these conditions are not fulfilled. In addition, many references are quite old or are represented by monographs or PhD Theses, which could not be found and investigated by me or/and readers.
Based on my comments, I recommend authors to select some national-level journal.
Reviewer 2 Report
Your hypothesis and experimental design is straightforward but not clearly written: I to compare the effect of fencing in vegetation under grazing and human harvesting along the period of observation. Descriptors for i) each of the individual species differences in density, cover and biovolume (cylinder, what you call size index) and ii) overall vegetation descriptors: diversity, which is breakable in evenness and richness. Your hypothesis (H1) is that there are differences in the descriptors between the set of fenced and non-fenced plots. So, Please state your research question clearly. The sentence in lines 64 to 66 is too vague.
line 90 - Plots are not the same size. Please explain why? Where they established by a minimal-area criteria so that they are appropriate to differences in richness in each habitat type?? What is their geometrical form? square? rectangular. Are the non-fenced and fenced plots in pairs in the same habitat conditions? At a short distance from each other?
Size index is better described as 'biovolume' in literature. line 114
line 115 - Species richness is an objective number for each plot (fence). What do you calculate a richness average for? Sorry, I really do not understand this.
line 116 - the sentence is meaningless. Please see the definition of 'species turnover' in literature. Turnover is the rate of change in species. Please make this clear.
line 117 - The Shannon-Weaver Index combines two components: richness and evenness. It is not clear how you separate the evenness component from the shannon-weaver index alone. Please explain this better.
line 152-154 - You should describe the overall pattern of difference in species cover (or density) in fenced and non-fenced, not say that 'some chamaephyte species have significant differences. The fact is that the large majority of species do not have significant differences as the t-tests have a probability of p > 0.05.
line 172 and 1821, 182 - please explain 'progressive values' This means that you associate the position in DCA to successional progressive status?
188-197 - dominance-diversity and comparison with a McArthur model is not referred in methods, please do this. Besides the interpretation that you make (192-197) is not supported by your results and seems speculative. Please delete this.
Table 3 - I find this table superfluous to be in the text. Table 2 would be enough. Move this to supplementary material. 'other relevant data' is vague. Please explain concrete data in the table.
lines 246-250 improvement seems expectable, but most of your t values (p>> 0.05) say the contrary (authors say this themselves in the analysis of results). Explain what improved (woody chamaephytes?) and what did not, was irrelevant or fluctuated randomly. Moreover, how do you evaluate improvement? Diversity is greater? There are more protected species? Or is the present vegetation associated with more mature successional stages? Please state this precisely.
264-274 - I find this irrelevant to your results discussion. I advise authors suppress this.
277 -279 the same , this paragraph is very confusing. What is being discussed here are factors that were not studied by authors. E.g. irrigation. So , what is the relevance of putting it to cmaparison with your results?
280-285 this is the most relevant conclusion of your study. Why not emphasize this instead of saying it 'en passant' in the middle of a paragraph?
285-286: you lack a reference to justify this as an explanation, as this does not derive from your results.
297 -296 - I would be careful about saying your study shows overall phytomass increased with fencing. Probably it did, but what you analyzed was each of the individual species. I do not see the total cover variations compared by a t- student means test anywhere. Either you do this test or you use expressions like ' suggests overall phtyomass increased' etc.
297 -312 the explanation you hypothesize could be right. But you did not study any of this (litter, number of surviving buds, dung concentration in plots, etc.), so state clearly that this paragraph contains hypothesis to explain the observed changes.
311-312 - Move this to results.
314 -317 this is repeating what is in the introduction and abstract, not a conclusion. I recommend deleting this.
321-322 - Irrelevant because you did not study climatic effects, so this is speculative. Do not use 'it is clear'. You could say that some variation in vegetation in the observed period might be attributable to factors that were not controlled by authors, for instance, climatic factors. But, in general, one´s results always can be explained by otter factors. First, you assume you did not control for such factors and you still think they could have been relevant to explain results. Are you sure you want to say this in your article? Second: the overall improvement of vegetation is irrelevant to your study hypothesis (vegetation will be different inside and outside fences). Namely, the climate will affect both fenced and non-fenced equally, So, why writing this here?
I suggest suppressing the whole discussion on climate and the reasons for improvement that were not studied.
As general comment is that something of environmental variables: climate, indicators of human and animal pressure, physiography, soil type etc. could have been measured and put in the statistical analysis, for instance with CCA (Canonical Correspondence Analysis). The relevant environmental and exploration controls would have been sorted out and analysed in detail. In concrete, a BACI (Before and After Control Impact) design could have been used. Therefore part of the explanation hypothesis you discuss could have been more supported.
Reviewer 3 Report
Overall this is a well written manuscript which is thoroughly and appropriately analysed. My main comments are
1) We need some information about how the ‘outside’ transects were chosen, as well as the supplied details about how the enclosures were placed. At the moment this is not mentioned. Were the outside transects matched to the enclosures and how, was the same total area surveyed inside and outside? Figure 1 should show the locations of the outside transects as well as the enclosures. It is also unclear how many line-intercepts were surveyed per enclosure/outside (could give n in the tables).
2) DCA is not the most robust of the multivariate methods, the second axis is a kind of artefact. Suggest to redo the ordination with NMDS to check the result. I am also not clear why the two panels are needed or which transects and species are included in each of the panels. Suggest presenting a simple NMDS of species composition across all the inside and outside transects, coloured in open and black as now, with the arrows to show the paired samples if they are paired. This would show more clearly any overall differences in species composition between inside and outside.
Some line comments below:
Discussion
Lines 182-184: Thus, the floristic heterogeneity has decreased after six years of protection. It is likely that the cessation of overgrazing and other human impacts increased the diversity inside the enclosures.
These two sentences are not immediately easy to reconcile and seem contradictory. Could there be some elaboration or rewriting of this section? I think the first is an interpretation of the DCA figure, but I cannot understand how this figure supports this interpretation because the figure is not clear to me. Axis 1 does not seem to distinguish the inside and outside transects particularly – seems that some other source of environmental variation is more significant than enclosure?
Paragraph about dominance diversity curves from line 188 – how did you fit these curves and what are the supporting statistics to accompany it? What software did you use? Should be included in the methods and supported with the statistics (the inside and outside curves look a very similar shape to each other to me, but happy to be overruled by supplied data).
Line 250, any interpretation for the deterioration of some enclosures?
Line 262, more sustainable in terms of what? Local income? Plant size? Species richness? Line 252, improvement in terms of what?
Table 2.
Column ‘enclosure number’ -> Number of enclosures. Would be clearer.
Caption: Descending order of what? More instructive to sort by cover as in 3 and 4?
Can you add n as well as number of enclosures? I am unsure how many transects were conducted inside each enclosure, and how many outside are in the comparisons. Is each t-test balanced? NB this is also a lot of t-tests… is some correction applied? Perhaps a % change like your RID would be just as useful.
You could use the p values to highlight (in bold, or with asterisk, or at the top of the table) the species which have shown significant change. Or replace the t-tests and sort by relative change %)
Your autocorrect has produced Stipagrostis ciliate in the species list (ciliata)
Table 3 - Why cover % values missing towards the end of the table?
Table 4 – if these species are only found outside the enclosures, what is the number of enclosures column signifying? Do you mean the number of outside transects they were found in?
Why are the bottom half of Cover % data missing? Put <0.5% if that’s the case?
Table 5&6
Relative number -> % of total spp
Figure 3. significant change using which table? Table 2?
Fig 4. I don’t understand the separation into two panels. Why not ordinate all the data together? In DCA the axis 2 is a sort of artefact – suggest redoing this with NMDS if you want to interpret both axes. Ordination of enclosures according to importance value data (cover) -> ‘using cover % as the measure of abundance’ would be clearer.
Conclusions line 322, can you say a bit more about the climatic variation over the last 6 years? It is referred to (also in abstract) but is never elaborated. Has it been exceptionally wet or dry over the study period, even anecdotally? Also though, would these climatic differences not generally affect both the inside and the outside equally? Or is there a serious suggestion that changes in rainfall should affect the inside and outside differently, in which case did you consider this in the reserve design?
Reviewer 4 Report
Revision report of the manuscript entitled: Protection of mountainous vegetation against human impacts in South Sinai, Egypt
The manuscript is very interesting and well written in every section, It is represent an important contribution to the studies on relationships between grassland ecology and management.